# Attend First, Consolidate Later:
# On the Importance of Attention in Different LLM Layers

**Amit Ben Artzy** and **Roy Schwartz**

{amit.benartzy,roy.schwartz1}@mail.huji.ac.il

## Abstract

In decoder-based LLMs, the representation of a given token at a certain layer serves two purposes: as input to the attention mechanism of the current token; and as input to the attention mechanism of future tokens. In this work, we show that the importance of the latter role might be overestimated for some layers. To show that, we start by manipulating the representations of previous tokens; e.g., by replacing the hidden states at some layer $k$ with random vectors (Fig. 1). Our experimenting with four LLMs and four tasks show that this operation often leads to small to negligible drop in performance. Importantly, this happens if the manipulation occurs in the top part of the model—$k$ is in the final 30–50% of the layers. In contrast, doing the same manipulation in earlier layers can lead to chance level performance. We continue by switching the hidden state of certain tokens with hidden states of other tokens from another prompt; e.g., replacing the word "*Italy*" with "*France*" in "What is the capital of *Italy*?". We find that when applying this switch in the top 1/3 of the model, the model ignores it (answering "*Rome*"). However if we apply it before, the model conforms to the switch ("*Paris*"). Our results hint at a two stage process in transformer-based LLMs: the first part gathers input from previous tokens, while the second mainly processes that information internally.

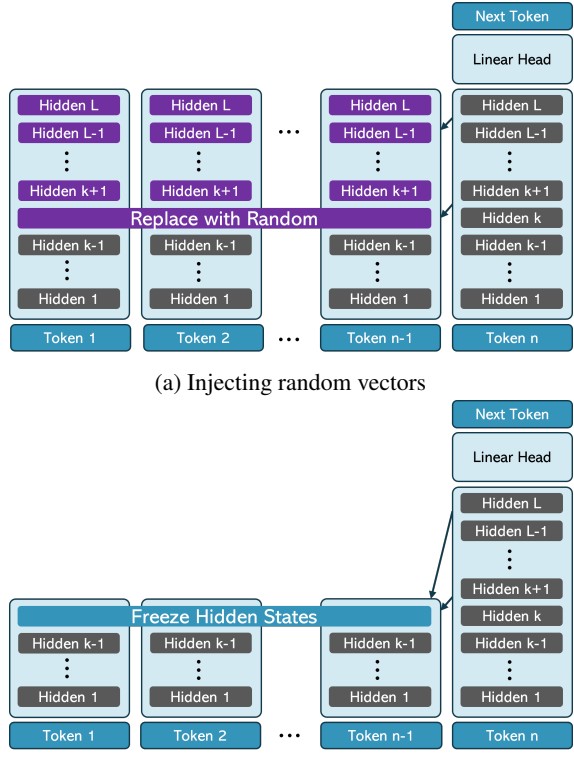

(a) Injecting random vectors

(b) Freeze Generation

Figure 1: To evaluate the role of previous hidden states as input to the attention mechanism, we devise two setups: (a) we replace the hidden state at layer $k$ with a random vector, and use it as input to layer $k + 1$, which continues processing as normal; (b) starting from a given layer $k + 1$, the hidden representations of previous tokens are frozen, and the attention mechanism attends to their hidden states at layer $k$.

## 1 Introduction

The attention mechanism in transformer-based (Vaswani et al., 2017) LLMs allows information to flow from the hidden representation of one token to another. While this process is the same for all model layers, previous work has shown that different layers capture different types of information (Niu et al., 2022; Geva et al., 2020, 2022; Press et al., 2019; van Aken et al., 2019) It is therefore not entirely clear that this flow of information is equally important for all layers. Can we find a distinction between layers that aggregate information from previous tokens, and those that process this information internally?

To better understand these dynamics, we apply various manipulations to the hidden states of all tokens barring the current one, and evaluate their impact on the model's performance over various tasks. We consider several different manipulations,

e.g., replacing the hidden state at layer $k$ with random vectors; and replacing the hidden states of all upper layers ($\ell > k$) with those of the last unmanipulated layer ($k$). We note that none of the manipulations in this work involves further training or fine-tuning.

We experiment with four LLMs (Llama2-7B, Touvron et al., 2023; Mistral-7B, Jiang et al., 2023; Yi-6B, Ai et al., 2024; and Llemma-7B, Azerbayev et al., 2023) across four tasks. Our results show that transformers are surprisingly robust to manipulations of their previous tokens. Freezing up to 50% of the layers results in some cases in no loss in performance across multiple tasks. Moreover, replacing up to 30% of the top layers with random vectors also results in little to no decrease. Importantly, we identify a distinct point where LLMs become robust to these manipulations: applying them at that point or later leads to little to moderate drop in performance, while doing it earlier leads to a drastic drop in performance.

To further study this phenomenon, we consider a third manipulation: switching the hidden states of certain tokens with a hidden state computed based on a separate prompt. E.g., in factual question answering tasks ("What is the capital of *Italy*?"), we replace the hidden state of the token "*Italy*" with that of the token "*France*" from another prompt. Our results are striking: in accordance with our previous results, doing this manipulation at the top 1/3 of the model leads to no change in prediction. However, doing it earlier leads to the generation of the output corresponding to the change ("*Paris*" instead of "*Rome*").

Finally, we consider dropping the attention mechanism altogether, by skipping the attention block in all layers starting a given layer $k$. As before, we find in some cases a high variance in how important attention mechanism is across layers: doing this at the bottom layers leads to severe performance degradation, while doing it at higher layers results in smaller drops, and even matches baseline performance in some tasks.

Our results shed light on the way information is processed in transformer LLMs. In particular, they suggest a two-phase process: in the first, the model extracts information from previous tokens. In this phase any change to their hidden representation leads to substantial degradation in performance. In the second phase, information is processed internally, and the representation of previous tokens matters less. They also have potential implications for

making transformer LLMs more efficient, by allowing both to skip upper attention layers, and accordingly, reducing the memory load of caching these computations. We publicly released our code.[1]

## 2 Manipulated LLM Generation

Decoder-only transformers consist of a series of transformer blocks. Each block contains an attention block and a feed-forward block.[2] Formally, to generate token $n + 1$, we process the $n$'th token by attending to all previous tokens $i \leq n$. Formally, for layer $\ell$, we define the attention scores $A_n^\ell$ as follows:

$$A_n^\ell = softmax(\frac{q_n^\ell \cdot K_n^\ell}{\sqrt{d}}) \cdot V_n^\ell$$

where

$$q_n^\ell = W_q^\ell \cdot x_n^{\ell-1}; \ K_n^\ell = W_k^\ell \cdot X_{1,...,n}^{\ell-1}; \ V_n^\ell = W_v^\ell \cdot X_{1,...,n}^{\ell-1}$$

In this setup, $W_{q/k/v}^\ell$ are weight matrices, $d$ is the hidden size dimension, $x_n^{\ell-1}$ is the representation of the current token in the previous layer, and $X_{1,...,n}^{\ell-1}$ is a matrix of the hidden representation of all tokens from the previous layer.

We highlight two important properties of transformers. First, the $K_n^\ell$ and $V_n^\ell$ matrices are the only components in the transformer block that observe the previous tokens in the document. Second, all transformer layers are defined exactly the same, as described above. In this work we suggest that perhaps this uniformity across layers needs to be reconsidered.

We aim to ask how sensitive a model is to observing the exact history tokens, or in other words—how much will observing manipulated versions of them impact it. Below we describe the manipulations we employ. We stress that all the manipulations in this work operate on the pre-trained model, and do not include any training or fine-tuning. See Fig. 1 for visualization of the different approaches.

### 2.1 Noise

First, we ask whether the content of the hidden state even matters. To do so, we replace the hidden states at layer $k$ of all previous tokens ($X_{1,...,n-1}^k$) with *random vectors*. The next layer ($k + 1$) gets these random vectors as input, and the following layers continue as normal. We use two policies

---

[1] `github.com/schwartz-lab-NLP/`
`Attend-First-Consolidate-Later`

[2] We omit normalization and residuals for brevity.

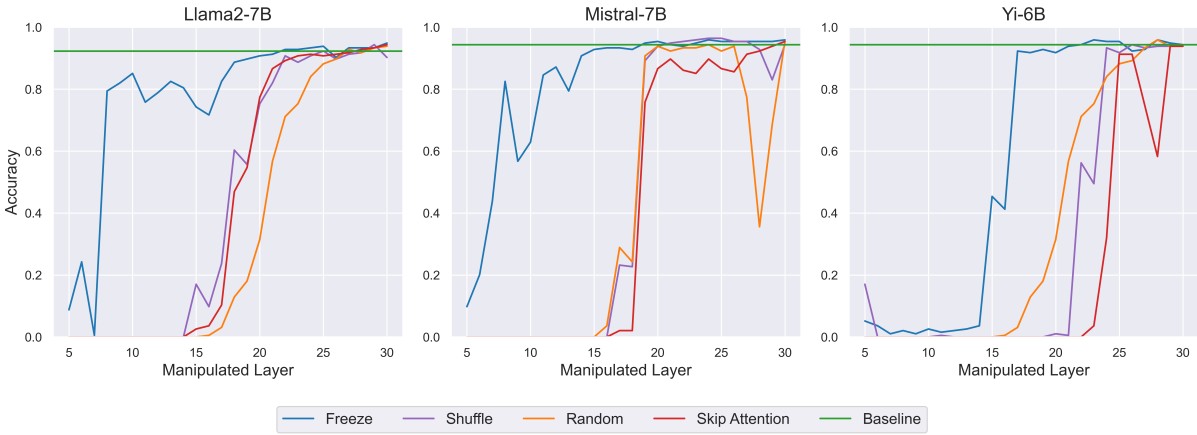

Figure 2: Manipulating the history tokens of different LLMs on the **capitals** dataset across different layers. We observe that all models become robust to the freeze manipulation after about 15 layers ($\approx 50\%$ of the layers), and to the other manipulations after about 20–25 layers.

for introducing noise to the history of tokens, both ensuring the noisy hidden states have the same norm as the original hidden states: **shuffle**, where we take the original hidden state and shuffle its indexes in a random permutation; and **random** where we randomize a vector of variance 1 and mean 0 then re-scale it such that the norm would be the same as the original hidden state.

If the model is successful in this setup at some layer, we may conclude that it has already gained the majority of the relevant information from previous layers, and in practice it is not making excessive use of this information in higher layers.

## 2.2 Freezing

We next turn to address the question of whether deep processing of the previous tokens is needed. To do so, we freeze the model at layer $k$ and copy the hidden state at that layer to all subsequent layers.[3] Formally, for $\ell > k$, we set $X^\ell_{1,\dots,n-1} = X^k_{1,\dots,n-1}$. If this manipulation would result in large performance degradation, we may conclude that subsequent processing in higher layers is important. Alternatively, a minor to no drop in performance would indicate that perhaps the processing up to layer $k$ is sufficient.

## 3 Experimental Setup

### 3.1 Datasets

We aim to understand the flow of information in various test cases. For this purpose, we consider four

benchmarks described below: two that we curate, which allow us to perform nuanced, meticulous manipulations; and two other benchmarks for standard tasks—question answering and summarization.

**Capitals**   We devise a simple fact-extraction QA dataset, which consists of 194 country-capital pairs. The dataset is in the format of "What is the capital of X?". To align the model to return the capital only, rather than a full sentence such as "The capital of X is Y", we use a 1-shot setting. We report exact match accuracy.

**Math exercises**   We compile a second dataset, consisting of simple math exercises of addition and subtraction of single digit numbers. We consider two variants: 2-term (i.e., the subtraction/addition of two numbers, e.g., "1 + 2 ="), and 3-term (e.g., "1+2+3 ="). In both cases, we only consider cases where the answer is also a single (non-negative) digit. In 3-term, we also verify that each mid-step can be represented as a single token. This results in 110 2-term instances, and 1210 3-term instances.

This dataset has a few desired properties. First, it has a clear answer, which facilitates evaluation. Second, perhaps surprisingly, it is not trivial—the math-tailored LLM we experiment with for this task (Llemma) only reaches $\approx 80\%$ on 2-term and $\approx 50\%$ on 3-term. Third, it allows us to easily increase the level of difficulty of the problem (by considering 2-term vs. 3-term). We report exact match accuracy.

**SQuAD**   We also consider the Stanford Question Answering Dataset (SQuAD; Rajpurkar et al., 2016), a dataset consisting of question-paragraph

---

[3]This is common practice in early-exit setups (Schuster et al., 2022), where some tokens require deeper processing than the previous ones that performed an early-exit.

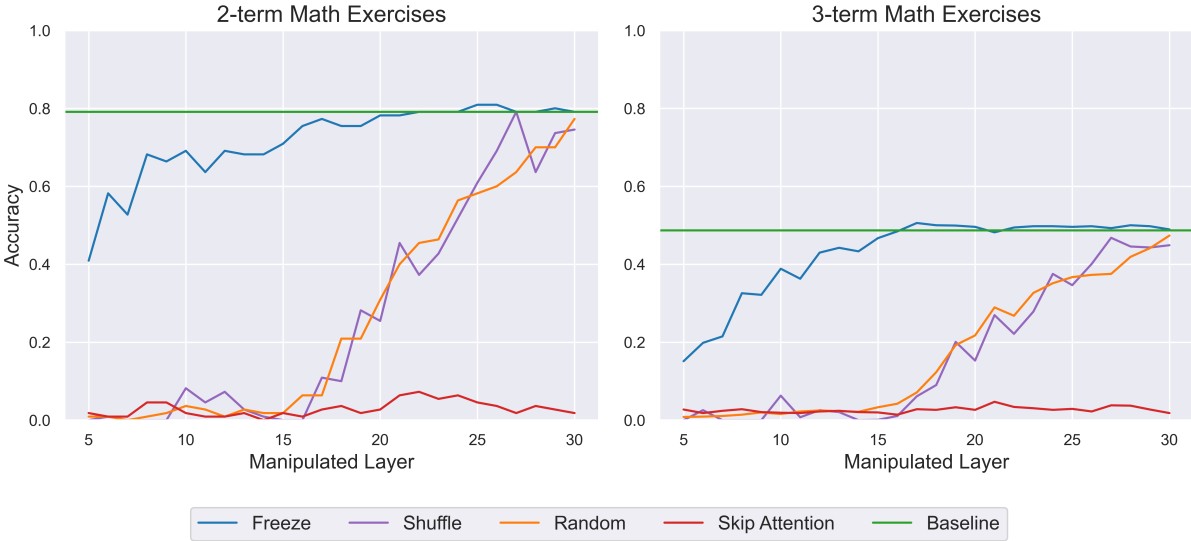

Figure 3: Manipulation results on both versions of the **math exercises** dataset with the Llemma model. The model is highly resilient to the freeze manipulation starting layer 16 in both cases, while far less robust to the other manipulations.

pairs, where one of the sentences in the paragraph contains the answer to the corresponding question. The task is to correctly output the segment that contains the answer. We sample 1,000 instances from the SQuAD test set and report exact match.

**CNN/Daily Mail**   This dataset (Hermann et al., 2015) contains news articles from CNN and the Daily Mail. The task is to generate a summary of these articles. We sample 100 instances from the CNN/Daily Mail test set and report averaged rouge-1 and rouge-l scores (Lin, 2004; Papineni et al., 2002).

### 3.2   Models

For the textual tasks (Capitals, SQuAD, and CNN/Daily Mail), we experiment with three open-source, decoder-only models, each containing 32 layers: Llama2-7B (Touvron et al., 2023), Mistral-7B (Jiang et al., 2023), and Yi-6B (Ai et al., 2024).

For the math exercises dataset, we observe that these models perform strikingly low, so we experiment with Llemma-7B (Azerbayev et al., 2023), a Code Llama (Rozière et al., 2023) based model finetuned for math.

### 4   Results

**Capitals**   Figure 2 shows the results of our manipulations on the capitals dataset. We first note that all LLMs are surprisingly robust to the different manipulations. When freezing the top $\approx 50\%$ of

the model, all models reach similar performance as the baseline (unmanipulated model). For the noise manipulations, we observe the same trend, though at later layers: For both manipulations, the model matches the baseline performance if applied at the top $\approx 30\%$ of the model.

We also note that, interestingly, in almost all cases we observe a critical layer $k'$, for which the model performs almost at chance level if manipulated in layers $i \leq k'$, while substantially improving if applied afterwards. While this point varies between models and manipulation types, e.g., from layer 8 (Llama2-7B, freeze) to 25 (Yi, shuffle), the phenomenon in general is quite robust.

Our results hint that LLMs exhibit a two-phase processing: the first part gathers information from previous tokens. At this part the content of previous hidden states is highly important. In contrast, at the second part, the model mostly consolidates this information, and is far less sensitive to such manipulations.

**Math exercises**   We next consider the math exercises dataset (Fig. 3). Again, we observe that freezing the top 1/2 of the model results in a similar performance to the baseline in both setups (2-term and 3-term). However, here the shuffle and random manipulations perform similar to the baseline only when applied at the top 10% of the model. Here we also observe that we do not see a clear transition point from chance level performance to

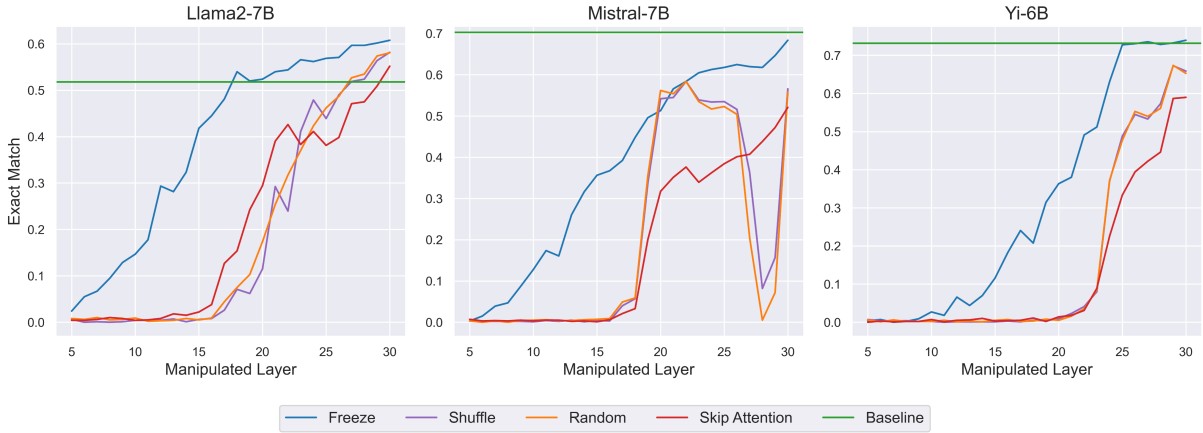

Figure 4: The effect of different manipulations on the **SQuAD** dataset. The Llama2-7B model is resilient to all manipulations after 18 (freeze) to 27 (skip attention) layers. Interestingly, results improve over the baseline if these manipulation are applied later. Yi is resilient to freezing (25 layers), though not to the other manipulations. Mistral is not resilient to any manipulation.

baseline-level, but rather a more steady increase. Interestingly, the trends in the 2-term and 3-term settings are similar; despite the fact that the 2-term problems are substantially easier to the model.

**SQuAD** We now turn to consider common NLP tasks, and start with SQuAD (Fig. 4). We first observe for two of the three LLMs, applying the freeze manipulation leads to the same trend as before: comparable (or even better!) results as the baseline. This happens starting layer 20 (Llama2-7B) or 25 (Yi). In contrast, for Mistral, the manipulated model only matches the performance of the full model after 30 (of 32) layers.

Considering the two noise manipulations, we observe that for Llama2-7B, both variants also reach the baseline performance after 25–27 layers. However, the other two models never fully reach it. Nonetheless, we note that for these models, we clearly observe the transition point observed in the capitals experiments: Between 15–23 layers, model performance is at chance level, and afterwards it dramatically improves. These results further support the two-phase setup.

**CNN / Daily Mail** Finally, we consider the CNN/ Daily Mail dataset (Fig. 5)

Yi-6B matches baseline performance at top layers across all manipulations. As in SQuAD, using the freeze manipulation, Llama2-7B performs similar or even slightly better than the baseline if applied in the final ≈30% of the layers. Results for Mistral-7B with the freeze manipulations are close, though clearly inferior to the baseline. In contrast, the two noise manipulations lead to substantially

lower scores in Llama2-7B and Mistral-7B.

**Discussion** Our results demonstrate a few interesting trends. First, we observe that in almost all cases, models are robust to freezing, in some cases as early as after 50% of the layers. We also note that in some cases (capitals, SQuAD with Llama2-7B) LLMs are robust to adding noise, but in general this does lead to noticeable performance degradation. Nonetheless, we still observe a rather consistent phenomenon with these manipulation, which shows that applying them too early leads to chance-level performance, and at a certain layer, results suddenly improve dramatically (albeit not reaching the baseline performance).

Our results suggest two-step phase in the processing of LLMs: a first step that gathers information from previous tokens, and a second that consolidates it. We next turn to further explore this hypothesis, by presenting two additional manipulations—replacing the hidden representation with that of another token from a different prompt; and skipping the attention mechanism altogether.

## 5 Injecting Information from a Different Prompt

To further test the two-phase hypothesis, we study the impact of "injecting" new information to the model, in the form of replacing the hidden representation of a given token with that of another token from a different prompt.[4] We experimented

---

[4]This process is often called "patching" (Hendel et al., 2023; Ghandeharioun et al., 2024).

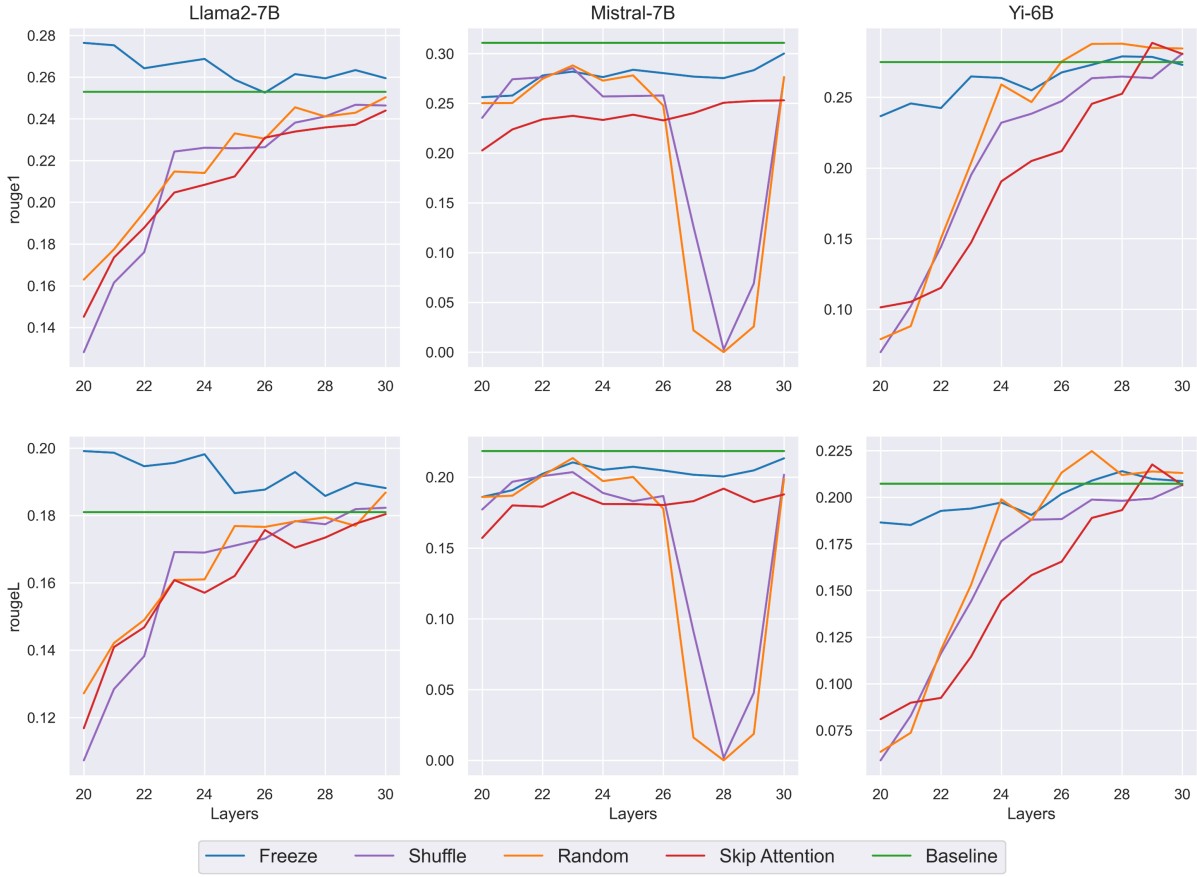

Figure 5: The effect of different manipulations on LLM performance on the **CNN/Daily Mail** dataset. Yi-6B reaches baseline performance at top layers. As before, Llama2-7B is resilient to freezing starting layer 20. Other models reach similar performance, but still inferior to the baseline. All models are not resilient to the other manipulations.

with the capitals dataset. For example, given the question "What is the capital of *Italy*?", we replace the hidden states corresponding to the word *"Italy"* in layer $\ell$ with the hidden states corresponding to *"France"* at layer $\ell$ from another prompt. We randomized pairs of countries which are represented with the same number of tokens.

Our results are shown in Fig. 6. For each model, we identify a clear transition point: before it, the model answer conforms to the patched value (e.g., "*Paris*" in the example above), and afterwards the model is unimpacted by the manipulation, returning the original answer ("*Rome*"). These results further illustrate the two phases we observed in previous experiments.

## 6 Is Attention Needed at Top Layers?

Our results so far indicate that the role of previous tokens is far more important in the bottom layers of the model than in top ones. A question that arises now is whether the attention mechanism is even needed in those top layers.

To study this question, we experiment with skipping the attention block in those layers, and only applying the feed-forward sub-layer.[5] We find that in three of the four datasets (capitals, Fig. 2; SQuAD, Fig. 4; and CNN / Daily Mail, Fig. 5), the effect of this process is similar to that of the shuffle manipulation. I.e., in some (though not all) cases the models are surprisingly robust to this process.

In contrast, and perhaps surprisingly, we find that skipping the attention block in the math exercises dataset leads to chance-level performance in all cases (Fig. 3). This might indicate the nature of this problem, where each and every token is critical to give the final answer, forces the model to use the attention mechanism all the way through.

## 7 Related Work

To better understand the inner-working transformers, previous work has explored the roles of the different transformer layers. Niu et al. (2022)

[5] Following preliminary experiments, we also skip the normalization prior to the attention.

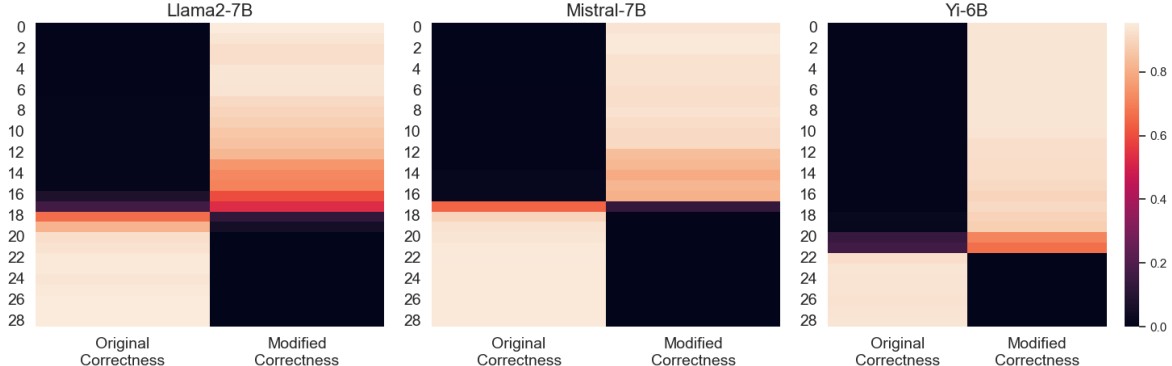

Figure 6: The effect of injecting information from a different prompt (E.g., replacing "*Italy*" with "*France*" in "What is the capital of *Italy*?"). All models exhibit a clear two-phase behavior: in the first part, the injection changes the output to be the modified answer (e.g., "Paris"). In the second, the model is unaffected by the injection (answering "Rome").

found that linguistics features can be extracted from hidden representations. Geva et al. (2020, 2022) demonstrated that lower layers are associated with shallow patterns while higher layers are associated with semantic ones.

Press et al. (2019) and Mandava et al. (2020) have trained from scratch transformers with different sub-layer ordering and found some variants that outperform the default ordering. Related to our work, they observed that oreederings that with more attention layers at the bottom half and more feed-forward at the top tended to perform better, hinting that the attention is more important at the bottom layers.

Early exit methods (Schwartz et al., 2020; Xin et al., 2020; Schuster et al., 2022; Elhoushi et al., 2024), which speed up LLMs by processing only the bottom part of the model, also provide evidence that the top layers of the model have already gained relevant information from previous tokens.

Prior work tried to better understand the information encapsulated in hidden representations. Hendel et al. (2023) demonstrated that patching an operator (e.g., the "→" token) from in-context tasks to another context preserves the operation. Ghandeharioun et al. (2024) showed that patching can be seen as a generalization of various prior interpretability methods and demonstrated how this method can be used in other cases, e.g., feature extraction. Both of these works aimed to study the hidden representations in transformers and the features they encode. In contrast, we propose to patch different vectors into some context to learn more about the flow of information between the tokens

in context.

Previous work also experimented with manipulating LLMs. Meng et al. (2022) corrupted hidden states to identify and edit specific neurons responsible to specific outputs. The concurrent work of Lad et al. (2024) investigated swapping layers to better understand the process of token generating, and hypothesized about stages of inference. Gromov et al. (2024) explored deleting layers for pruning purposes. We, however, utilized different manipulations to better understand the flow of information in LLMs.

## 8 Conclusion

We investigated the role of the attention mechanism across a range of layers. We applied various manipulations over the hidden states of previous tokens, and showed that their impact is far less pronounced when applied to the top 30–50% of the model. Moreover, we switched the hidden states of specific tokens with hidden states of other tokens from another prompt. We found that there is a distinct point, at the top $1/3$ of the model, where before it the model conforms to the switch, and afterwards it ignores it, answering the original question. Finally, we experimented with dropping the attention component altogether starting from a given layer. We found again, that in some cases (though not all), doing this at the top 30% of the model leads to a small effect, while much larger earlier.

Our results shed light on the inner workings of transformer LLMs, by hinting at a two-phase setup of their text generation process: first, they aggregate information from previous tokens, and then

they decipher the meaning and generate new token. Our work could further be potentially extended to reduce LLMs costs: First by skipping the attention component in upper layer, and second by alleviating the need to cache their output for future generations.

## Acknowledgements

This work was supported in part by NSF-BSF grant 2020793.

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
