# OpenReview forum: "Attend First, Consolidate Later: On the Importance of Attention in Different LLM Layers"
_EMNLP/2024/Workshop/BlackBoxNLP — BlackboxNLP 2024_

### Official Review · Reviewer_t1gc · 2024-08-28

**Overall Assessment:** 3
**Confidence:** 3

**Best Paper:**

1

**Best Paper Justification:**

-

**Comments Questions Suggestions And Typos:**

citations mentioned above:

@inproceedings{NEURIPS2019_2c601ad9,
 author = {Michel, Paul and Levy, Omer and Neubig, Graham},
 booktitle = {Advances in Neural Information Processing Systems},
 editor = {H. Wallach and H. Larochelle and A. Beygelzimer and F. d\textquotesingle Alch\'{e}-Buc and E. Fox and R. Garnett},
 pages = {},
 publisher = {Curran Associates, Inc.},
 title = {Are Sixteen Heads Really Better than One?},
 url = {https://proceedings.neurips.cc/paper_files/paper/2019/file/2c601ad9d2ff9bc8b282670cdd54f69f-Paper.pdf},
 volume = {32},
 year = {2019}
}

@inproceedings{voita-etal-2019-analyzing,
    title = "Analyzing Multi-Head Self-Attention: Specialized Heads Do the Heavy Lifting, the Rest Can Be Pruned",
    author = "Voita, Elena  and
      Talbot, David  and
      Moiseev, Fedor  and
      Sennrich, Rico  and
      Titov, Ivan",
    editor = "Korhonen, Anna  and
      Traum, David  and
      M{\`a}rquez, Llu{\'\i}s",
    booktitle = "Proceedings of the 57th Annual Meeting of the Association for Computational Linguistics",
    month = jul,
    year = "2019",
    address = "Florence, Italy",
    publisher = "Association for Computational Linguistics",
    url = "https://aclanthology.org/P19-1580",
    doi = "10.18653/v1/P19-1580",
    pages = "5797--5808",
}



@article{meng2022locating,
  title={Locating and Editing Factual Associations in {GPT}},
  author={Kevin Meng and David Bau and Alex Andonian and Yonatan Belinkov},
  journal={Advances in Neural Information Processing Systems},
  volume={36},
  year={2022},
  note={arXiv:2202.05262}
}

**Paper Summary:**

In this paper, the input to the attention mechanism at different layers is manipulated: previous hidden states are replaced with random vectors, "frozen" hidden representations from previous layers, or representations of counterfactual tokens. In experiments across 3 textual tasks and with 3 models, as well as a math task with two models, authors find that models are relatively robust to this noise in higher layers, while performing at chance level if corruption happens in earlier layers. The paper concludes that the results are evidence of a two-stage process in Transformers, with a first phase where information is gathered from previous tokens, and a second phase where representations are mainly refined internally. These findings could motivate pruning of attention heads in higher layers.

**Summary Of Strengths:**

- the paper applies careful manipulations of previous token representations, and shows convincingly that model behavior can change abruptly depending on which layers is injected with noisy/counterfactual information. The point is verified with different noise types and skipping attention completely.
- the paper is well-written.
- the paper fits the workshop well.

**Summary Of Weaknesses:**

- There are two relevant strands of research that are not discussed in related work, and also reduce the novelty of findings.
  - One is research on pruning attention heads (e.g. Michel et al. 2019, Voita et al. 2019), which has already shown that many Transformer attention heads have little relevance on a prediction. This work argues that attention is more important in earlier layers than later ones (mirroring Press et al. 2019).
  - previous interpretability work, for example the Causal Mediation Analysis as used in Meng et al. (2022), works with a similar principle (applying corruption at some point and measuring the effect), and has been used to arrive at similar conclusions (specific hidden states can be decisive for the prediction, and we can localize and edit these decisive states to manipulate the prediction).
- The analysis in this paper is relatively coarse-grained, especially in relation to work mentioned in the previous point. For example, this paper corrupts/skips all attention in top layers, and concludes that attention in top layers is less important and may be skippable, but this conclusion seems to broad. For example, Mistral-7B is sensitive to corruptions in layer 27, and Llemma is sensitive to corruption in all layers. I'd suggest moving to a more fine-grained analysis, but this will likely make the work more similar to Meng et al. (2022).

---

### Official Review · Reviewer_r2LQ · 2024-09-09

**Overall Assessment:** 4
**Confidence:** 4

**Best Paper:**

1

**Best Paper Justification:**

N/A

**Comments Questions Suggestions And Typos:**

- Missing citations

Stages of inference:

[Vedang et al., 2024, The Remarkable Robustness of LLMs: Stages of Inference?](https://arxiv.org/abs/2406.19384)

Referring to activation pathing:

[Geiger et al., 2020, Neural Natural Language Inference Models Partially Embed Theories of Lexical Entailment and Negation](https://arxiv.org/abs/2004.14623)

[Vig et al., 2020, Investigating Gender Bias in Language Models Using Causal Mediation Analysis](https://proceedings.neurips.cc/paper/2020/hash/92650b2e92217715fe312e6fa7b90d82-Abstract.html)

Typos

- Figured are referenced in the text just with their number, instead of "Figure X".
- Figure 2, caption: "... observe that the all" -> "... observe that all"
- line 278 "of even better" -> "or even better"

**Paper Summary:**

The paper studies the importance of layers when modifying internal representations of the LLMs. The study uses four LLMs (Llama2-7B, Mistral-7B, Yi-6B, and Llemma-7B) across four tasks: a custom capitals dataset, math exercises, SQuAD, and CNN/Daily Mail summarization. By using two types of perturbations to the previous tokens -replacing the original representations by random vectors, and freezing the intermediate representations by copying the same hidden state as input to all subsequent layers- the authors demonstrate models are highly robust to these changes in the top 30-50% of the layers, suggesting a two-phase processing stage, a first step that gathers information from previous tokens, and a second that consolidates it. To further test this hypothesis, two more experiments are proposed, performing activation patching on the residual stream, and skipping attention in in upper layers, finding supporting evidence.

**Summary Of Strengths:**

- The paper is easy to follow, and the different experimental settings are well explained.
- The paper proposes a comprehensive study, featuring multiple LLMs (Llama2-7B, Mistral-7B, Yi-6B, and Llemma-7B) and datasets (capitals dataset, math exercises, SQuAD, and CNN/Daily Mail summarization).
- Results look consistent across models, datasets and types of interventions, providing strong evidence for the two-stage processing hypothesis.

**Summary Of Weaknesses:**

- The different stages in the forward-pass, and the apparently ineffectiveness of later layers have been proposed in previous work (Vedang et al., 204, Gromov et al., 2024, Elhousi et al., 2024), making the findings in this paper less novel. However, here the focus is on the context utilization.

- I missed a more detailed analysis into what the final layers are doing, and in which cases they are relevant.


[Gromov et al., 2024, The Unreasonable Ineffectiveness of the Deeper Layers](https://arxiv.org/abs/2403.17887)

[Vedang et al., 2024, The Remarkable Robustness of LLMs: Stages of Inference?](https://arxiv.org/abs/2406.19384)

[Elhousi et al., 2024, LayerSkip: Enabling Early Exit Inference and Self-Speculative Decoding](https://arxiv.org/abs/2404.16710)

---

### Official Review · Reviewer_YoZ4 · 2024-09-09

**Overall Assessment:** 4
**Confidence:** 4

**Best Paper:**

1

**Best Paper Justification:**

No comments

**Comments Questions Suggestions And Typos:**

Suggestions/questions:
- As mentioned above, it would be interesting to see the difference between immediate early exiting and freezing only the hidden states but not the attention.
- The model seems to be able to recover from replacing the hidden states past a certain layer. Is it a possibility that after that layer, the model is able to ignore noise and recover the previous hidden states from its 'residual stream'? In this case, the model may still be needing previous tokens in those later layers. Would replacing/randomizing the hidden states of all layers l > k for some k (similar to the freezing technique) be a viable experiment?
- What could specifically be happening around layer 28 of Mistral?

Minor suggestions/clarity:
- Section 5: It would be good to mention whether this experiment is performed on all pairs of capitals.
- L060: 'and two additional tasks we compile', it would be clearer to simply list the tasks or at least what the tasks are roughly about.

**Paper Summary:**

The paper aims to shed light on which components of large decoder Transformer models are necessary for next-token prediction.
It does so by 1) freezing or replacing hidden states with noise (by shuffling/generating random ones) of all previous tokens in certain layers or 2) skipping the attention mechanism in a certain layer.
These methods are applied to multiple models applied to two question answering tasks, a summarization task, and a simple arithmetic task.

For the majority of the tasks and models investigated, the drop in performance after these edits is small to negligible in the final ~30% of layers, indicating that the models have acquired all task relevant information from earlier tokens in early layers, and do not need to pay as much attention to these historical tokens in the later layers to predict the correct next token.
All models are more robust to freezing hidden states than to randomizing hidden states. Sometimes, freezing the hidden states even leads to better predictions than letting the model run in full.
For the mathematical arithmetic task, the model is not as robust to randomization of the hidden states, and not robust at all to skipping the attention mechanism.

Generally, the steep increase in accuracy on natural language tasks after a certain layer suggests that these decoder Transformer models operate in two phases to solve these tasks: first acquiring relevant information from previous tokens, and then after that layer, consolidating this information internally. There is no such steep increase in accuracy for the mathematical task.

**Summary Of Strengths:**

The paper is well written.
The paper discusses a variety of tasks, models, and types of internal manipulations.
There are some interesting results, for example, differences between tasks (arithmetic vs. natural language: this raises some interesting questions about how the attention mechanism is used to learn algorithmic tasks).

**Summary Of Weaknesses:**

It was not entirely clear to me whether the "freeze" approach is actually equivalent to early exiting (This seems to be implied in line 388 when comparing to Schuster 2022) or freezing *only* the hidden states, but not the attention mechanism in the later layers (Which seems to be implied in section L165).
I think this is an important distinction to make, and might be interesting to compare: if early exiting, which is more widely studied, works just as well -- what is there to gain by only skipping the attention components / only skipping the hidden state components of later layers? In other words: how much 'internal processing' or consolidation is necessary? The paper currently does not answer this question.


The paper's engagement with other related interpretability work is somewhat limited:
- There is literature on early exiting/early decoding and interpretability, such as the LogitLens (nostalgebraist 2020) and papers using LogitLens to interpret intermediate layers (e.g. Merullo et al. 2023, which also experiments with a capital city prediction task throughout the layers, Halawi et al. 2023, which also notes that intermediate layers can achieve better accuracy in some cases).
- There is also previous work on intervening on early hidden states by replacing them with those of another input (eg. 'interchange interventions' for encoder Transformers, Geiger et al. 2021).
- The only future work suggestion listed is for improving efficiency, but what could these findings mean for future *interpretability* work?

Because the methods of replacing/freezing states are quite general, and because of the two weaknesses listed above, the paper and its findings may not be novel: a similar paper with more interaction with previous work would be preferable. My assessment of the paper is therefore somewhere between ambivalent and worthy. Still, I have rounded my overall assessment score up to a 4, since the paper is clear, well written, and leads to some interesting general questions.

---

### Decision · Program_Chairs · 2024-09-20

**Decision:**

Accept

**Comment:**

While the reviewers all agree that the paper presents an exciting setup with interesting findings, a large amount of missing related work is pointed out that the authors should include in a revised version of the paper. Since the topic that is tackled by the paper has been covered in earlier work as well, it is highly important the authors make clear how their work differentiates from these papers.